# Embedding Biomimetic Magnetic Nanoparticles Coupled with Peptide AS-48 into PLGA to Treat Intracellular Pathogens

**DOI:** 10.3390/pharmaceutics14122744

**Published:** 2022-12-08

**Authors:** Salvatore Calogero Gaglio, Ylenia Jabalera, Manuel Montalbán-López, Ana Cristina Millán-Placer, Marina Lázaro-Callejón, Mercedes Maqueda, María Paz Carrasco-Jimenez, Alejandro Laso, José A. Aínsa, Guillermo R. Iglesias, Massimiliano Perduca, Concepción Jiménez López

**Affiliations:** 1Department of Biotechnology, University of Verona, Strada Le Grazie 15, 37134 Verona, Italy; 2Department of Microbiology, Faculty of Sciences, University of Granada, 18071 Granada, Spain; 3Departamento de Microbiología, Pediatría, Radiología y Salud Publica (Facultad de Medicina) & BIFI, Universidad de Zaragoza, 50009 Zaragoza, Spain; 4CIBER de Enfermedades Respiratorias (CIBERES), Instituto de Salud Carlos III, 28029 Madrid, Spain; 5Department of Applied Physics and Instituto de Investigación Biosanitaria ibs. GRANADA, NanoMag Laboratory, University of Granada, 18071 Granada, Spain; 6Department of Biochemistry and Molecular Biology I, University of Granada, 18071 Granada, Spain

**Keywords:** BMNPs, PLGA, AS-48, photothermia, hyperthermia, magnetic nanoparticles, antimicrobial peptide, monocytes, *Mycobacterium tuberculosis*

## Abstract

Among the strategies employed to overcome the development of multidrug-resistant bacteria, directed chemotherapy combined with local therapies (e.g., magnetic hyperthermia) has gained great interest. A nano-assembly coupling the antimicrobial peptide AS-48 to biomimetic magnetic nanoparticles (AS-48-BMNPs) was demonstrated to have potent bactericidal effects on both Gram-positive and Gram-negative bacteria when the antimicrobial activity of the peptide was combined with magnetic hyperthermia. Nevertheless, intracellular pathogens remain challenging due to the difficulty of the drug reaching the bacterium. Thus, improving the cellular uptake of the nanocarrier is crucial for the success of the treatment. In the present study, we demonstrate the embedding cellular uptake of the original nano-assembly into THP-1, reducing the toxicity of AS-48 toward healthy THP-1 cells. We optimized the design of PLGA[AS-48-BMNPs] in terms of size, colloidal stability, and hyperthermia activity (either magnetic or photothermal). The stability of the nano-formulation at physiological pH values was evaluated by studying the AS-48 release at this pH value. The influence of pH and hyperthermia on the AS-48 release from the nano-formulation was also studied. These results show a slower AS-48 release from PLGA[AS-48-BMNPs] compared to previous nano-formulations, which could make this new nano-formulation suitable for longer extended treatments of intracellular pathogens. PLGA[AS-48-BMNPs] are internalized in THP-1 cells where AS-48 is liberated slowly, which may be useful to treat diseases and prevent infection caused by intracellular pathogens. The treatment will be more efficient combined with hyperthermia or photothermia.

## 1. Introduction

Directed chemotherapy to treat local infections is not only necessary to avoid undesired secondary effects, but also to reduce the number of drugs needed to control the infection. This advantage becomes crucial in the context of antibiotics. In fact, the World Health Organization (WHO) is warning about the increase in the failure of common antibacterial therapies due to the development and selection of multidrug-resistant (MDR) bacteria, mainly due to the generalized and/or extended drug administration [1]. As an example, MDR *Mycobacterium tuberculosis* strains have developed resistance to the two major “first-line” drugs used against tuberculosis (isoniazid and rifampicin). Treatment of MDR *M. tuberculosis* requires long-lasting therapy (administered daily for 7–9 months), based on the use of less efficient and more toxic drugs [2].

Among the strategies to overcome the problem of MDR pathogens, directed chemotherapy combined with another local therapy (e.g., magnetic hyperthermia) has gained great interest. Wu et al. [3] showed the potential of combining a photothermic–chemical treatment for a murine subcutaneous abscess induced by methicillin-resistant *Staphylococcus aureus*. Wang et al. [4] developed a dual-targeted platform for synergistic chemo–photothermal therapy against multidrug-resistant Gram-negative bacteria and their biofilms. More recently, Jabalera et al. [5] designed the AS-48-BMNP nano-assembly, where biomimetic magnetic nanoparticles (BMNPs) were simultaneously used as drug carriers and hyperthermia agents, and the circular antimicrobial peptide AS-48 was used as the active antimicrobial compound. The nano-assembly showed strong bactericidal effects on the planktonic cultures of both Gram-positive and Gram-negative bacteria, with the effects being remarkable against the Gram-negative species *(Pseudomonas aeruginosa* and *Klebsiella pneumoniae),* which are naturally resistant to AS-48. Notably, BMNP nano-assemblies can be magnetically guided and/or concentrated at the target [6].

The therapeutic approach is more challenging when the disease is caused by intracellular bacteria, such as *M. tuberculosis* [7]. This pathogen reaches the lungs of infected individuals, where it is phagocyted by alveolar macrophages forming intracellular acidic vesicles named phagosomes. In the majority of cases, *M. tuberculosis* prevents phagosome maturation and, hence, escapes from eradication by the host cell and turns the phagosome into a niche for cell replication. Infected macrophages stimulate the immune system, triggering the formation of cellular superstructures, the granulomas, which contain the infected macrophages and prevent antituberculosis drugs from reaching the pathogen [2]. To successfully treat this kind of pathogens by a magnetically directed therapy, the availing of a carrier that can penetrate granulomas and locally increase drug bioavailability is crucial. Therefore, for the nano-formulation, the already designed AS-48-BMNPs have to, e.g., carry the bacteriocin, allow magnetic driving, ensure stability at the physiological pH, release the bacteriocin at the target, and behave as hyperthermia agents. However, this nano-formulation needs to be optimized to enhance cellular uptake, reduce the toxicity of AS-48 for non-infected cells, and ensure intracellular drug release.

MamC-mediated biomimetic magnetic nanoparticles have proven to be suitable nanocarriers for directed combined therapy [5,8]. These BMNPs are chemically produced by introducing a magnetosome protein from *Magnetococcus marinus* MC-1, MamC (expressed as a recombinant protein), in the reaction mixture from which magnetite nucleates grow. MamC chelates Fe cations and acts as a template for magnetite nucleation, producing magnetic nanoparticles in different sizes, surface charges, and magnetic properties from those chemically produced (MNPs). In terms of size, BMNPs are larger (~35–40 nm) than MNPs (<20 nm), which allows BMNPs to be superparamagnetic and to display a larger magnetic moment per particle than that displayed by MNPs. This characteristic ensures an enhanced magnetic response once an external magnetic field is applied for guidance and/or concentration, and favors a non-magnetic behavior of BMNPs in the absence of an external magnetic field, which prevents BMNP aggregation [6].

In terms of surface charge, the advantage of using BMNPs versus MNPs is mainly related to the greater ease in the process of functionalization. MNPs usually have an isoelectric point close to 7; thus, they need to be coated to successfully bind the relevant molecule and to keep the nano-assembly at the physiological pH stable [9,10]. Instead, in the case of BMNPs, MamC confers to the nanoparticle-surface functional groups, which switch their isoelectric points to ~4.5 [8]. This is important since, as the nanoparticle is negatively charged at the physiological pH, drug-binding by electrostatic interaction and nano-assembly stability at this pH value are ensured, drug release is triggered in acidic environments as BMNPs become uncharged [8], and BMNPs behave as stimulus-response drug delivery systems. Furthermore, BMNPs are hyperthermia agents that are able to locally raise the temperature to the hyperthermia therapeutic range (41–45 °C) following the application of an alternating magnetic field (AMF) [11,12], which further increases the drug release at the target [13,14,15].

Novel compounds with unexploited mechanisms of action are required to deal with bacterial infections. Among them, the bacteriocin AS-48 is a cationic 70 residues-long head-to-tail cyclized peptide produced by *Enterococcus faecalis*, with a potent bactericidal effect on Gram-positive bacteria [16,17,18], including several human pathogens [19,20]. AS-48 is very stable in a broad range of conditions (pH, temperature, salt concentrations) and resistant to most proteases. This antimicrobial peptide in solution is arranged in two different dimeric forms as a function of the physicochemical environment: dimeric form I (DF-I), in which the molecules are linked by hydrophobic interactions (hydro-soluble form), and dimeric form II (DF-II), in which the molecules are linked by hydrophilic interactions (hydrophobic form). Basically, the bactericidal effect is electrostatic dependent and is exerted by the accumulation of DF-I on the negatively charged membrane, followed by dissociation into DF-II and insertion into the lipid bilayer, which causes the creation of pores [21]. The antibacterial action is based on a receptor-independent mechanism, thus reducing the development of stable and transmissible resistance [19]. Additionally, AS-48 has not shown any remarkable toxicity in preclinical studies, indicating its safety at preventing and treating infections, even those caused by MDR microorganisms [22]. However, Gram-negative bacteria show naturally increased resistance to AS-48 due to the presence of the outer membrane. The nano-assembly AS-48-BMNPs combine the thermal and mechanical damages of the outer membrane induced by the rotation of the BMNPs (under an AMF stimulus), in such a way that the bacteriocin can reach the cytoplasmic membrane. In these conditions, AS-48 can exert antimicrobial effects even against microorganisms considered naturally resistant [5].

Despite the advances made to date, improving the cellular uptake of the nanocarriers is crucial to reach an intracellular pathogen. In this scenario, Vurro et al. and Jabalera et al. [12,23] tested the PLGA encapsulation of BMNPs to enhance their internalization in U87MG and HepG2 cells, respectively. Polylactic-co-glycolic acid (PLGA) is a copolymer of the ester family, widely used to prepare nano- and microparticles for nanomedicine purposes [24,25,26] when biocompatibility and biodegradability are required. Moreover, PLGA has been approved for human therapies by the Food and Drug Administration (FDA) and the European Medicines Agency (EMA) [27,28]. The biocompatibility for human cells is strictly related to its spontaneous hydrolysis, which leads to the release of lactic and glycolic acid monomers that can be easily metabolized by cells through the Krebs cycle [26]. Nevertheless, while embedding compounds, such as those reported in [12], is relatively easy, the embedding of AS-48 is not trivial due to its ability to insert and induce permeability in lipid membranes [29]. In fact, previous attempts to encapsulate AS-48 in liposomes yielded non-functional formulations. Therefore, being able to encapsulate a nano-formulation containing AS-48 means that the direction of this molecule is a real step forward in the potential use of this bacteriocin.

The present study focuses on the design of a nano-formulation that allows the combination of directed chemotherapy and local hyperthermia and that could be potentially used against intracellular pathogens. Several nano-formulations based on AS-48-BMNP were optimized and characterized to improve cell uptake, reduce their toxicity in healthy (non-infected) cells, and delay bacteriocin release until the nano-formulation is inside the host. As a cell model, human macrophages (THP-1) are used since these cells are usually invaded by *M. tuberculosis*. This could be the first step toward a new approach in the treatment of diseases caused by intracellular pathogenic species, even if belonging to MDR strains, using a safe nanomaterial.

## 2. Materials and Methods

Several nanoformulations have been designed and characterized in the present study, which are summarized in Table 1.

### 2.1. Magnetic Nanoparticles: Synthesis and Functionalization

Biomimetic magnetic nanoparticles were synthesized by the protein-mediated co-precipitation method under anaerobic conditions (Table 1) following the protocol described in Valverde-Tercedor et al., 2015 [30]. MamC was produced and purified as a recombinant protein following a previously described protocol [30]. Briefly, transformed *Escherichia coli* TOP10 cells (Life Technologies: Invitrogen, Grand Island, NY, USA) were grown at 37 °C in Luria–Bertani (LB) broth supplemented with ampicillin, and the production of the MamC protein was induced with isopropyl-β-d-thiogalactopyranoside (IPTG, Fisher BioReagents, Pittsburgh, PA, USA). Once produced, the purification of the protein was carried out under denaturing conditions by fast protein liquid chromatography (FPLC, GE Healthcare) using immobilized metal affinity chromatography (IMAC, GE Healthcare, Chicago, IL, USA). Fractions containing MamC were refolded at 4 °C through dialysis in suitable buffers.

The synthesis of BMNPs was carried out at 1 atm of total pressure and 25 °C from oxygen-free solutions, at a pH value of 9, following standard protocols [8]. All experiments were conducted under anaerobic conditions inside an anaerobic Coy chamber (96% N_2_/4% H_2_, Coy Laboratory Products, Grass Lake, MI, USA). Samples were incubated for 30 days, and then the solids, following magnetic concentration and discarding of the supernatant, were washed with deoxygenated Milli-Q water. The precipitated solid was stored in HEPES buffer (HEPES 10 mM, NaCl 150 mM, pH 7.4) and sterilized.

AS-48 was purified from cultures of the enterococcal UGRA10 strain [31] on Esprion 300 plus glucose (1%) (DMV Int., Veghel, Netherland) in a pH-controlled device following the conditions established by Ananou et al. [32]. Briefly, the fermentation supernatant was purified by cationic exchange chromatography on a Sepharose Big Beads resin (GE Amershan) and the eluted fractions were applied to a hydrophobic C18 column, to finally be purified to homogeneity by reversed-phase, high-performance liquid chromatography (RP-HPLC) on a Vydac 218TP510 semipreparative column (the Separation Group, Hesperia, CA, USA). The concentration of purified derivatives, determined by measuring UV absorption at 280 nm in a Nanodrop, was converted to a protein concentration using the molecular extinction coefficient.

To produce AS-48-BMNPs nano-assemblies, 5 mg of nanoparticles were mixed with 1 mL of aqueous 100 μM AS-48 in HEPES buffer (10 mM HEPES, 150 mM NaCl, pH 7.4), and the samples were incubated for 24 h [5]. After the incubation time, the nano-assemblies were collected with a magnet, and pellets were carefully washed twice with 1 mL HEPES buffer. The supernatants were measured by UV-Vis spectroscopy at 280 nm and the non-adsorbed AS-48 was estimated using a calibration line and subtracted from the initial concentration suspension. Thus, the coupling efficiency was expressed as the percentage of AS-48 bound (AS-48(%)_bound_) and calculated by the following equation, where AS-48_fed_ is the amount of the peptide used for the coupling, while AS-48_lost_ is the unbound amount quantified in the waste supernatants. Finally_,_ the total bound amount was expressed per mg of magnetite.
$$\mathrm{AS}-48{\left(\%\right)}_{\mathrm{bound}}=\frac{\left(\mathrm{AS}-48_{\mathrm{fed}}-\mathrm{AS}-48_{\mathrm{lost}}\right)}{\mathrm{AS}-48_{\mathrm{fed}}}\times 100\;$$

### 2.2. PLGA Encapsulation Protocol

The synthesis of the PLGA nano-formulation embedding BMNPs was archived using the single emulsion–evaporation method (Table 1) [26,33]. To avoid the aggregation phenomenon, we tested different concentrations of PVA (0.1–2%), and to reduce the loss of AS-48, we performed several trials varying the PLGA concentration (2.5–20 mg/mL); the BMNP concentration was kept constant (5 mg/mL). Additionally, the influences of different organic phases (acetone, acetonitrile, dimethyl sulfate, acetone/ethanol) on the nano-assembly formation were tested. Thus, the following protocol was optimized to achieve a stable hyperthermia agent within the nanometric range. Briefly, 10 mg of PLGA copolymer (in the presence of 5 mg of BMNPs or AS-48-BMNPs) was dissolved in 1 mL of organic solution (85% acetone and 15% ethanol) and dripped into 10 mL of polyvinyl alcohol (PVA) 1% aqueous solution under sonication. The obtained emulsion was left under shaking overnight to enhance the organic solvent evaporation. Finally, the nano-assemblies were collected using a magnet and washed 3 times with PBS pH 7.4 before being resuspended in the desired buffer.

Empty PLGA nanoparticles were prepared with the same procedure, just avoiding the presence of the payloads. In this case, PLGA nanoparticles were collected by centrifugation (15,550× *g* for 20 min) (Eppendorf Centrifuge 5804R) and washed three times with PBS. The final pellet was re-suspended in 1 mL of sterile PBS buffer and stored at 4 °C. In the case of the synthesis of the AS-48-BMNPs, all supernatants were measured by UV-Vis spectroscopy at 280 nm to check for the presence of residual amounts of AS-48, using the same calibration curve previously prepared. Thus, the encapsulation efficiency was expressed as the percentage of the bacteriocin loaded (AS-48(%)_loaded_) and calculated by the following general equation, where AS-48_fed_ stands for the quantity bound to BMNPs and is used for the encapsulation process, while AS-48_lost_ stands for the free AS-48 in the waste supernatants
$$\mathrm{AS}-48{\left(\%\right)}_{\mathrm{loaded}}=\frac{\left(\mathrm{AS}-48_{\mathrm{fed}}-\mathrm{AS}-48_{\mathrm{lost}}\right)}{\mathrm{AS}-48_{\mathrm{fed}}}\times 100$$

### 2.3. Physicochemical Characterization of the Nano-Formulations

The nano-formulations obtained were characterized in terms of size (Nanoparticle Tracking Analysis–NTA- and Atomic Force Microscopy–AFM), surface charge (ζ-potential), and surface chemistry (Fourier Transformed Infra-Red, FT-IR) AFM is a scanning probe microscopy (SPM) based on atomic force interactions between the sample and a nanometric probe; the technique offers information about morphology and surface topography. The analysis was performed in dry mode on a mica disk. The AFM output is a direct estimate of the nanoparticle diameter [34]. Thus, the AFM analysis, 20 μL drops of each previously sonicated sample, were deposited on 20 mm diameter mica discs. An NT-MDT Solver Promicroscope (Moscow, Russia), with a single-crystal silicon–antimony-doped probe and a gold-coated tip (NSG-01 from NT-MDT), was used to collect images. The microscope was calibrated using a calibration grating (TGQ1 from NT-MDT) to reduce non-linearity and hysteresis in the measurements. The obtained images were processed using the program Gwyddion [35], and a statistical analysis as a function of the diameters on 35 different nanoparticles of each sample was conducted.

The NTA analysis is based on the dynamic light scattering phenomenon where the diffusion coefficient of each tracked particle was calculated in an aqueous environment by the Stokes–Einstein equation. Therefore, NTA analyses yield the hydrodynamic radii of the nano-formulations [36]. Analyses were performed from suspensions of the nano-formulations in PBS pH 7.4. The nanoparticle tracking analysis (NTA) was performed using a Malvern NanoSight NS300 instrument (Worcestershire, UK) on diluted samples (1:1000) at 25 °C. For each sample, 3 sequences of 30 s with 25 FPS were recorded. The data analysis was carried out using NTA 3.4 Build 3.4.003 software [36].

Nanoparticle tracking analysis (NTA) was performed using a Malvern NanoSight NS300 instrument (Worcestershire, UK) on diluted samples (1:1000) at 25 °C. For each sample, three sequences of 30 s with 25 FPS were recorded. The data analysis was carried out using NTA 3.4 Build 3.4.003 software [36].

The ζ-potential of the nano-formulations at pH 7.4 and 25 °C (10 mM NaClO_4_) was analyzed by using Nano Zeta Sizer ZS (ZEN3600, Malvern Instruments, Malvern, Worcestershire, UK). Before performing the measurements, samples were diluted 20 times obtaining a final concentration of BMNPs close to 0.25 mg/mL.

Lyophilized samples were characterized by an FTIR spectrometer (model 6600, Jasco, Japan) equipped with an attenuated total reflection (ATR) diamond crystal window (ATR ProOne). A total of 64 scans were collected in the wavenumber range from 4000 to 400 cm^−1^, at 2 cm^−1^ resolution.

### 2.4. Magnetic Properties

An AC generator was used to perform the magnetic hyperthermia experiments. The experimental setup consisted of induction heating coils made up of four turns of water-cooled copper, a power supply, and a chiller to maintain the temperature of the coils. Samples were analyzed at a fixed frequency of 120 kHz and under three magnetic field strengths, 13, 17, and 23 kA/m measured at the center of the coil, with an AC magnetic probe (NanoScience Laboratories Ltd., Staffordshire, UK). These parameters were chosen for the application of the alternating magnetic field since they are below the limit established by Hergt and Duzt in 2017 (Hf < 5 × 10^9^ Am^−1^s^−1^) [37] and were recently proposed by Herrero de la Parte et al. in 2022 [38] (Hf < 9.59 × 10^9^ Am^−1^s^−1^) as the biophysical limitations. All samples were previously thermostated at 37.0 ± 0.2 °C. The temperature increase as a function of time was measured with a fiber optic thermometer (Optocon AG, Dresden, Germany), and the specific absorption rate (SAR) and intrinsic loss power (ILP) of the different systems were calculated [39,40] using Equations (1) and (2).
(1)$$\mathrm{SAR}=\left(\frac{C\cdot V_{s}}{m}\right)\frac{\mathrm{dT}}{\mathrm{dt}}$$
(2)$$\mathrm{ILP}=\frac{\mathrm{SAR}}{{\mathrm{fH}}_{0}^{2}}$$
where C is the volumetric specific heat capacity of the sample (C_Water_ = 4185 J/LK), vs. is the sample volume (0.2 mL in the reported experiments), and m is the mass of solids in the sample (2 mg).

The photothermia experiments were performed in a microfuge tube (0.5 mL) containing 0.2 mL of suspension of the relevant nano-formulation in HEPES buffer. Nanoparticle concentrations were adjusted to [Fe] = 19 mM. During the experiments, each sample was irradiated from the top with a NIR laser (λ = 808 nm) at 0.5, 1, and 2 W/cm^2^ and visualized with a thermography camera (Flir 60 with 320 × 240 pixels, IR resolution, and thermal sensitivity <0.045 °C; FLIR Systems, Inc., Wilsonville, Oregon, USA), to measure temperature increases.

### 2.5. Stability and Bacteriocin Release

The releases of AS-48 bacteriocin from PLGA[AS-48-BMNPs] and AS-48-BMNPs were analyzed at different temperatures (4 °C, 20 °C, and 37 °C) and different pH values (7.4 and 5) by suspending the nano-assemblies in PBS and citrate buffers, respectively. The experiment was performed for one week and, at each specific time interval, the nano-assemblies were magnetically separated from the supernatant and resuspended in the fresh buffer to continue the time course experiment. The released bacteriocin was quantified from UV-Vis at 280 nm.

In addition, the release of AS-48 was identically analyzed following magnetic hyperthermia (frequency = 120 kHz, H = 23 kA/m, 120 min) or photothermia treatment (λ = 808 nm, at 1 W/cm^2^, 30 min). AS-48 release was indirectly analyzed over the time course experiment from the supernatants as detailed above. Each experiment was performed in triplicate for each condition.

### 2.6. Cell Culture

The human leukemia monocytic cell line, THP-1, was obtained from the European Collection of Animal Cell Cultures (Salisbury, UK). The cells were cultured in RPMI-1640 containing 10% heat-inactivated FBS supplemented with 2 mM L-glutamine, 100 U/mL penicillin, and 100 μg/mL streptomycin, in a humid atmosphere with 5% CO_2_ at 37 °C, and sub-cultured at a ratio of 1:10 once a week.

### 2.7. Cytotoxicity of the Nano-Assemblies

THP-1 cells were seeded onto 96-well black plates with clear bottoms (20,000 cells/well) in complete RPMI-1640 containing 10% FBS and were differentiated to macrophages adding 20 nM phorbol 12-myristate 13-acetate (PMA) to the medium for 48 h. Then, the culture medium was replaced with fresh medium/10% FBS and the different nano-assembly treatments were applied in a volume of 100 μL. Macrophages were incubated for 48 h in the absence (control) or in the presence of different nano-formulations: BMNP (229 μg/mL), AS-48 (32 μg/mL), PLGA (457 μg/mL), AS-48-BMNPs (229 μg/mL BMNPs and 32 μg/mL AS-48), PLGA[AS-48-BMNPs] (457 μg/mL PLGA, 229 μg/mL BMNPs and 32 μg/mL AS-48). To determine the cell viability, the resazurin assay was conducted, adding to each well 10 μL of 1 mM resazurin and determining the fluorescence at λex = 535/λem = 590 nm, in a microplate reader (HTX Microplate Reader BioTek Instruments, Winooski, VT, USA).

### 2.8. Cellular Uptake and Iron Content Estimation

To determine the amounts of internalized BMNPs in the different nano-formulations, THP-1 cells were seeded in 12-well plates (300,000 cells/well) and treated with BMNPs (229 μg/mL), AS-48-BMNPs (229 μg/mL BMNPs and 32 μg/mL AS-48), PLGA(BMNPS) (457 μg/mL PLGA, 229 μg/mL BMNPs), and PLGA[AS-48-BMNPs] (457 μg/mL PLGA, 229 μg/mL BMNPs and 32 μg/mL AS-48). In all cases, the final concentration of BMNPs was 229 μg/mL ([Fe] = 2.97 mM). Cells were incubated for 48 and 72 h and then trypsinized and transferred to 2 mL tubes and centrifuged at 8500× *g* for 5 min. Afterward, to dissolve the cell pellet with internalized nanoparticles, 100 μL of 37% HCl/10% H_2_O_2_ was added and maintained for 20 min at room temperature. Finally, 1 mL of 1% potassium thiocyanate in Milli-Q water was added to the tubing, and the absorbance at 490 nm was measured by UV–Vis spectroscopy. To obtain the percentage of endogenous iron (Fe_endogenous_), the following equation was applied: where Abs_Fe(internalized)_ is the absorbance collected by dissolving cell pellet, while Abs_Fe(fed)_ stands for the absorbance of the desired nano-formulation.
$${\mathrm{Fe}}_{\mathrm{endogenous}}\left(\%\right)=\frac{{\mathrm{Abs}}_{\mathrm{Fe}\;\left(\mathrm{internalized}\right)}}{{\mathrm{Abs}}_{\mathrm{Fe}\;\left(\mathrm{fed}\right)}}\times 100$$

### 2.9. Statistical Analysis

Statistical analyses were performed using Excel (16.65 20209) for Mac. For in vitro biological analysis, data represent means ± SEM of three independent experiments performed in triplicate, and statistical analyses were carried out using single-way ANOVA, with a Bonferroni’s post hoc test for the grouped analysis. Statistical differences between the treatments were considered significant at * *p* ≤ 0.05, ** *p* ≤ 0.001, *** *p* ≤ 0.0001.

## 3. Results and Discussion

### 3.1. Nano-Assemblies Physicochemical Characterization

BMNPs were previously characterized [30], and they consisted of pure stoichiometric magnetite with an average size of 39 ± 7 nm (sizes from 10 to 70 nm) and an isoelectric point (iep) of ~4.1. AS-48 was effectively coupled to BMNPs, yielding a load of 0.14 ± 0.04 mg AS-48/mg BMNPs. This load is consistent with that reported by Jabalera et al. [5]. Additionally, no AS-48 loss was observed during PLGA encapsulation. FTIR analyses were run to confirm the binding of AS-48, and the covering of the nano-assemblies with PLGA (Figure 1). The absorption bands at 800 and 900 cm^−1^ corresponded to the Fe-O bond of magnetite (Fe_3_O_4_) and were evident for both the BMNPs and the nano-assemblies, although the signal was less intense as the coating width increased (from AS-48-BMNPs) to PLGA[AS-48-BMNPs]). AS-48 and PLGA shared common adsorption bands within the region 1100–1500 cm^−1^. There were also distinctive bands for AS-48 at 801 and 837, 1543, 1656, 2936 cm^−1^ (C-C stretch, C=C bend, C=N bend, and O-H/C-H stretch, respectively). PLGA also showed distinctive bands at 862 and 1089, 1171, and 1752 cm^−1^ (C-C stretch, C-O stretch, and C=O bend). Many of these bands (1089, 1171, and 1752 cm^−1^) can also be detected in PLGA[AS-48-BMNPs], showing the existence of the coating. Furthermore, the presence of AS-48 in PLGA[AS-48-BMNPs] was directly demonstrated by SDS-PAGE (Figure 2), in which a clear band close to 7000 Da, corresponding to AS-48 (expected size 7149.5 Da), can be observed in lanes where AS-48 from a pure stock and PLGA[AS-48-BMNPs] were loaded.

The size of each nano-formulation was estimated by comparing two different techniques: nano-tracking analysis (NTA) and atomic force microscopy (AFM). NTA and AFM analyses of PLGA[BMNPs], PLGA[AS-48-BMNPs], AS-48-BMNPs, BMNPs, and empty PLGA nanoparticles, yielded mode average sizes below 120 nm (Table 2). The discrepancy in NTA and AFM data for PLGA nano-formulations is mainly associated with artifacts during AFM measurements related to the expansion of soft PLGA on the mica surface, with the consequent increase in size [33]. On the contrary, AFM data for harder materials, such as BMNPs, do not show such artifacts, with size data from both techniques (NTA and AFM) being comparable.

NTA size distributions of BMNPs (Figure 3a) showed several populations with sizes of 35, 63, 99, 149, and 229 nm, although the highest percentage of BMNPs was 99 nm. The average TEM size for the nanoparticles in this batch of BMNPs was 39 nm [30] and, therefore, it can be inferred that, although there were monodisperse BMNPs nanoparticles (NTA size of 35 nm), most BMNPs were aggregated in clusters of 2 to 3 BMNPs (NTA size of 99 nm). Regarding AS-48-BMNPs, most of the population exhibited sizes of 45 and 67 nm, although higher sizes of 131 and 181 nm were also detected. It is interesting to show that these data indicate that: (1) AS-48-BMNPs mostly consist of monodisperse populations of (non-aggregated) individually covered BMNPs, and (2) there are different sizes related to the covering of individual BMNPs (i.e., 45 and 67 nm). On the one hand, BMNP aggregation seems to be prevented in the presence of AS-48, providing AS-48-BMNP nano-formulations with higher colloidal stability (Figure 3b). Since BMNPs are superparamagnetic, the BMNP aggregation revealed in Figure 3a should be related to electrostatic and/or hydrophobic interactions between the nanoparticles, likely within the MamC domains present at the BMNP surface [8]. These forces change when AS-48 is present. As AS-48 is strongly positively charged at this pH [5], a preferential electrostatic interaction may occur between AS-48 and BMNP that destabilizes the former BMNP cluster. On the other hand, AS-48 attaches to BMNPs following a cooperative model [5], in which AS-48 binds to either the BMNP surface and/or to the previously attached AS-48 molecules. That leads to differences in the number of AS-48 molecules that are carried by a single BMNP, thus explaining the observed differences in the NTA population sizes.

PLGA[AS-48-BMNPs] preparation was optimized using PVA 1%, 10 mg of PLGA, 5 mg of BMNPs, and a mixture of acetone/ethanol with a ratio of 85/15. Indeed, lower concentrations (<1%) of the surfactant led to greater aggregation resulting in an increase in size; while a concentration greater than 1% did not bring more benefits. The polymer concentration used was the best one able to reduce the AS-48 loss while maintaining the nanometer size of the nano-assembly. Furthermore, among the different organic phases used (acetone, acetonitrile, dimethyl sulfate), only the mixture of acetone and ethanol did not negatively influence the binding between the peptide and BMNPs. Two well-defined populations with sizes of 115 and 155 nm (Figure 3c) were observed. This size range was adequate to ensure enhanced permeability and the retention (EPR) effect (<150 nm, [41]) and to allow an extended circulating time [42]. Again, the embedding in PLGA improves the colloidal stability of this nano-formulation, as the size ranges of the nano-formulations in Figure 3c are narrower than those of BMNPs and AS-48-BMNPs. Differences within the populations are likely caused by either the number of AS-48-BMNPs nano-assemblies enclosed in PLGA and/or the size of the enclosed individual AS48-BMNP nano-assemblies, related to differences in the number of the attached AS-48 molecules.

ζ-potential measurements of all nano-formulations at physiological pH show that they display negative surface charges (Table 3). For BMNPs, this negative charge is due to the carboxylic and hydroxyl groups present in MamC, which allow the electrostatic interactions with the cationic AS-48 through the amine groups of the peptide [5]. The binding of AS-48 to BMNPs blocks some of the negatively charged functional groups, thus becoming the surface less negative (Table 3, −32 mV for BMNPs, −15 mV for AS-48-BMNPs). Covering these nanoparticles with PLGA helps maintain the negative surface charges of the nano-assemblies, probably thanks to the carboxylic groups of the lactic acid in PLGA. In fact, PLGA[BMNPs] and PLGA[AS-48-BMNPs] show the ζ-potential of −19 ± 7 and −14 ± 6 mV, respectively (Table 3). Overall, the fact that all of the nano-formulations are charged at physiological pH helps improve colloidal stability, as their aggregation is prevented due to electrostatic repulsion. As pH decreases, the acidic functional groups protonate and the surfaces become less negative (Table 3). The roughly zero net charge (of the BMNPs) at pH = 5 is consistent with previous studies (Garcia-Rubia et al., 2018). Covering AS-48-BMNPs with PLGA confers the nano-formulations of net positive charges at acidic pH values, thus potentially favoring their interactions with the negatively charged cells without compromising the activity of AS-48, whose interactions with cell structures are most likely prevented until PLGA dissolution.

### 3.2. Nano-Formulations as Hyperthermia Agents

Magnetic Hyperthermia

All nano-formulations tested (BMNPs, PLGA[BMNPs], AS-48-BMNPs, and PLGA[AS-48-BMNPs]) were able to raise the temperature to the so-called therapeutic temperature (40–45 °C, transparent grey band) following their exposure to an alternating magnetic field (AMF) (Figure 4). Appendix A presents the summary of the specific absorption rate (SAR) and intrinsic loss power (ILP) for each nano-formulation. This temperature increase was dependent on the magnetic field strength of the AMF. The fastest and higher temperature increase occurred for BMNPs, for which in only ~15 s the therapeutic temperature was achieved, even at the lowest intensity of AMF (Figure 4a). As BMNPs were covered by a non-magnetic covering (i.e., AS-48 and/or PLGA), the ability to raise the temperature slightly decreased, although PLGA[AS-48-BMNPs] was still able to reach the therapeutic temperature within 30 s upon AMF exposure (23 kA/m, 120 kHz frequency). This shielding of the magnetic core by a non-magnetic coating was observed by numerous authors [43,44] and, in particular, for the TAT–PLGA[DOXO-BMNPs] nano-assembly by Jabalera et al. [12]. The nanoparticles inside the polymeric coating are mechanically constrained, leading to a reduction in the freedom of rotation, which results in an impediment to Brownian relaxation. Moreover, the constraint increases the chance of an interaction between the particles inside the PLGA which hampers the Néel relaxation. Both relaxation patterns are crucial for the magnetic hyperthermia performance [45,46,47].

### 3.3. Photothermia

As observed with magnetic hyperthermia, the nano-formulations tested were able to increase the temperature after a few seconds following laser exposure (λ = 808 nm) at a laser power density ranging from 1 to 2 W/cm^2^. Applying 0.5 W/cm^2^ was not enough to enhance the hyperthermia response. Again, the rate of the temperature increases and the final temperature reached were dependent on the laser power density. In almost all of the nano-formulations, the therapeutic temperature was only reached at ≤40 s following laser exposure at the highest laser power density (Figure 5). The attachment of AS-48 to the BMNPs decreases (but does not suppress) the ability of the nano-formulations to act as photothermic agents. This shielding of the organic coverage of BMNPs in terms of photothermal behavior was also observed by Jabalera et al. for TAT-PLGA (DOXO-BMNP) [5]. The FTIR analysis shows that AS-48 has a peak close to 808 cm^−1^ (801 cm^−1^); likely, the corresponding functional group (C-C) interferes with the photothermal performance by absorbing some of the radiation [46]. After the peptide’s absorption capacity is saturated, the heating rate increases linearly. Conversely, the PLGA polymer improves the response to hyperthermia (Figure 5a,b). As observed for the silica coating [46], PLGA could modify the thermal conductivity of the solution by improving the heating rate [48]. A summary of the calculation of the specific absorption rate (SAR) for all nano-formulations at different laser power densities is presented in Appendix A.

### 3.4. Cellular Uptake

One crucial characteristic in an efficient drug delivery nano-formulation is the ability to increase cellular internalization, which, in our case, is quantified as iron uptake, and then translated as the % of BMNPs internalized (Figure 6). While 42% (96.18 µg/mL) of the BMNPs incubated for 48 h were internalized in THP-1 cells (and 48% (109.92 µg/mL) at 72 h), higher percentages were obtained at 48 h for PLGA[BMNPs] and PLGA[AS-48-BMNPs] (56% (128.24 µg/mL) and 66% (151.14 µg/mL), respectively), and at 72 h (64% (146.56 µg/mL) and 78% (178.62 µg/mL), respectively). The slightly higher internalization values of PLGA[AS-48-BMNPs] were consistent with their positive ζ-potential, which enhanced the electrostatic interaction between this nano-formulation and the negative cellular membrane. These results are consistent with those of Vurro et al. and Jabalera et al. [12,23], in which BMNPs encapsulated in PLGA showed enhanced internalization in U87MG and HepG2 cells compared to non-encapsulated BMNPs, without affecting (in any case) cellular viability. PLGA likely promotes phagocytosis, overcoming BMNP internalization limits [49]. Figure 6b shows a brightfield microscopy picture after 72 h of treatment with PLGA[AS-48-BMNPs], in which no signs of cytotoxicity are shown in the cells.

### 3.5. Stability and AS-48 Release Pattern

Data in Figure 7 show that there is a release of AS-48 [AS-48 release (%)] from both nano-assemblies, AS-48-BMNPs and PLGA[AS-48-BMNPs], dependent on the pH, temperature, and the combination with hyperthermia. The release of AS-48 is favored at high temperatures for both pH values. In fact, at physiological pH (pH 7.4), the release percentage of AS-48 reached a maximum of 10%, 30%, and 45% for temperatures of 4, 20, and 37 °C, respectively, but when temperature increased to 45 °C, following laser application, the release of AS-48 reached ~60% (Figure 7b).

An identical trend was observed at pH = 5. The release percentage of AS-48 reached maximums of 20%, ~48%, and ~70% for temperatures of 4, 20, and 37 °C, and it reached 80% following laser exposure (Figure 7c,d). In all cases, the release of AS-48 from the nano-formulations was higher at acidic pH values compared to those of physiological pH values. On the one hand, pH = 5 is close to the isoelectric points of BMNPs, so the nanoparticles did not exhibit the negative charges at higher pH values, which were responsible for the electrostatic binding between the BMNPs and the cationic groups of AS-48 [12]. Once detached from BMNPs, the AS-48 released from the PLGA embedding is also favored at acidic pH values, as acidic environments trigger PLGA hydrolysis into its by-products, i.e., polylactic acid and polyglycolic acid [50].

Identically, thermal energy was previously shown to weaken the electrostatic bond and trigger a drug release [12]. Therefore, the lowest rate of the AS-48 release from the nano-formulations was observed at 4 °C (<10% in 7 days), while the faster and more intense release (~80%) occurred after only 30 min at acidic pH values under high temperatures induced by photothermia (45 °C). It is remarkable to note the difference in the time scale for AS-48 release between conditions with and without external stimuli, such as magnetic hyperthermia or photothermia. In the case of photothermia, 80% release occurs in the first 30, min and in magnetic hyperthermia, 50% release is reached in only 120 min compared to one week for the same release under non-hyperthermia conditions. Therefore, pH 7.4 and 4 °C should account for the storage and conservation of the nano-assemblies.

In summary, BMNP-based nano-assemblies can be magnetically concentrated at the target in less than 1 h (mice model, [6]). At physiological pH values, while being driven to the target, PLGA[AS-48-BMNPs] is stable (negligible AS-48 release, Figure 7a). Being positively charged, PLGA[AS-48-BMNPs] readily attach and are internalized by THP-1 cells (Figure 6a), likely by endocytosis [49]. At this point, environmental pH conditions change to more acidic environments [51,52], so the AS-48 release is triggered (Figure 7c). Moreover, once at the target, either magnetic hyperthermia or photothermia is applied, and the temperature increases inside the cell (Figure 5d) further enhances the AS-48 release in THP-1.

Interestingly, in terms of the AS-48 release rate, there is a trend observed in all experiments, such as the ability of PLGA to delay (in time) the AS-48 release. This result is important because the nano-formulation PLGA[AS-48-BMNPs] could be especially useful for longer treatments in which a delayed and lasting drug release is needed, for instance, to treat macrophages infected with *M. tuberculosis* [2].

### 3.6. Cytotoxicity of BMNP Nano-Formulations

Data in Figure 8 show that BMNPs and PLGA are cytocompatible for healthy (non-infected) THP-1 cells (>85% and >95%, respectively, cell survival after the treatment), while <3% THP-1 survival occurred following treatment with soluble AS-48 (32 μg/mL < MIC for *M. tuberculosis* [53]). These results highlight the importance of the direction of AS-48 since the administration of soluble AS-48 at these concentrations results in being highly toxic for healthy, non-infected, macrophages.

By coupling AS-48 to BMNPs, the viability of healthy macrophages increases up to 70%; furthermore, including AS-48-BMNPs in PLGA results in an increase in healthy THP-1 viability (up to 80%). This result is of particular interest; although PLGA[AS-48-BMNPs] shows the highest internalization, it also shows the lowest toxicity for healthy macrophages. This could be related to the slowest release rate of AS-48 (Figure 7).

Therefore, our data show that the nano-formulation PLGA[AS-48-BMNPs] could be a good candidate for the directed chemotherapy of AS-48, which could be combined with hyperthermia (magnetic hyperthermia and photothermia). This nano-formulation also enhances internalization and mediates a slow release of the bacteriocin suitable for extended antibacterial treatment, and reduces the mortality of non-infected cells with respect to those when treated with the free bacteriocin. Moreover, our hypothesis, which will be tested in the future, is that this nano-formulation allows non-infected macrophages to store AS-48 without affecting cell viability (for at least 72 h), likely preventing a spread if they are further infected by the pathogen.

## 4. Conclusions

The nano-formulations (AS-48-BMNPs, PLGA[AS-48-BMNPs]) are suitable in terms of size and surface charge, and because they behave as hyperthermia agents, to allow the directed chemotherapy of AS-48 combined with hyperthermia therapies (both magnetic hyperthermia and photothermia). In particular, PLGA[AS-48-BMNPs] shows the slowest AS-48 release at physiological pH values (<15% AS-48 release within the first 24 h), indicating a low chance of bacteriocin loss while magnetically concentrating at the target (~1 h in mice model) [6]. Moreover, this nano-formulation shows a high percentage of internalization in a model macrophage cell (>80%). The release of AS-48 from this nano-formulation is the slowest, which could be beneficial for extended treatments, but it can be accelerated up to 80% in minutes by combining with hyperthermia. Finally, this nano-formulation shows little cytotoxicity with respect to healthy (non-infected) cells, at least within the first 72 h. Therefore, little impact of this nano-formulation in healthy cells is expected while being guided/concentrated at the target. Furthermore, the internalization of these nano-formulations in healthy macrophages and the slow release of bacteriocin inside could prevent the pathogen from colonizing the cell and, thus, the spreading of the infection. Nevertheless, the present study only offers a proof of concept of a novel nano-formulation, whose performance needs to be tested in the future in a real intracellular infection model.

## Figures and Tables

**Figure 1 pharmaceutics-14-02744-f001:**
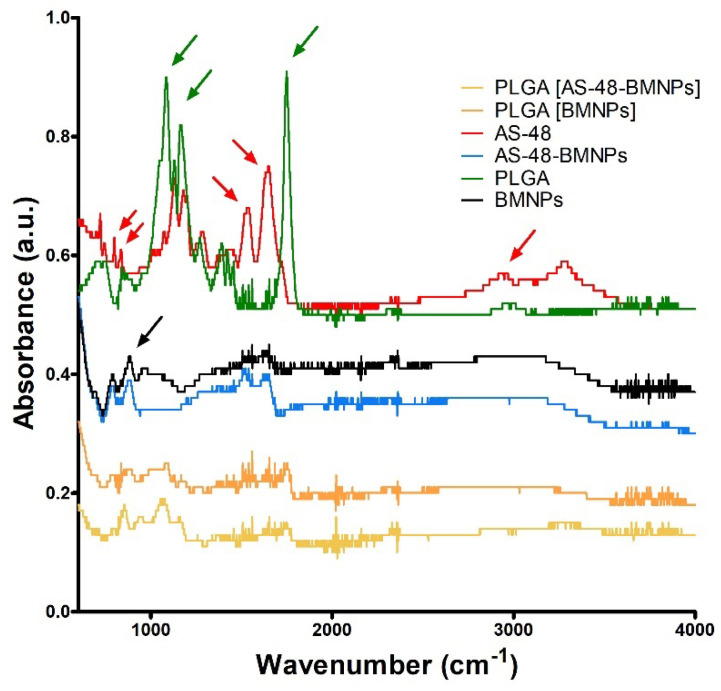
FTIR spectrum collected in the range between 600 and 4000 cm^−1^. The black arrow stands for adsorption bands associated with magnetite (lines in black), AS48 distinctive adsorption bands are marked with red arrows and those for PLGA are marked with green arrows.

**Figure 2 pharmaceutics-14-02744-f002:**
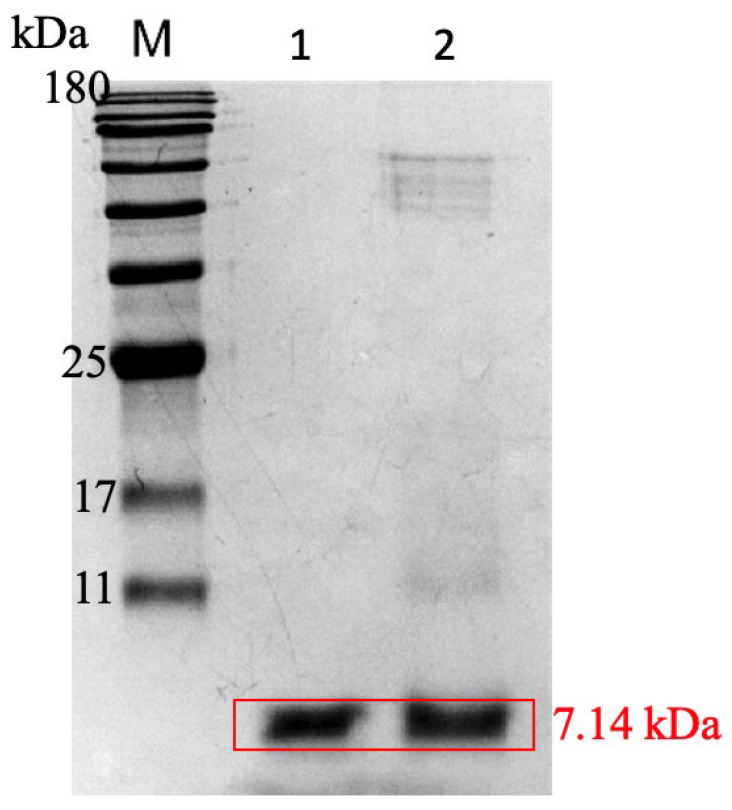
SDS-PAGE gel of the nano-formulation. Other than the marker (M), purified AS-48 (Lane 1) and PLGA[AS-48-BMNPs] (Lane 2) were loaded in the second and third lanes, respectively.

**Figure 3 pharmaceutics-14-02744-f003:**
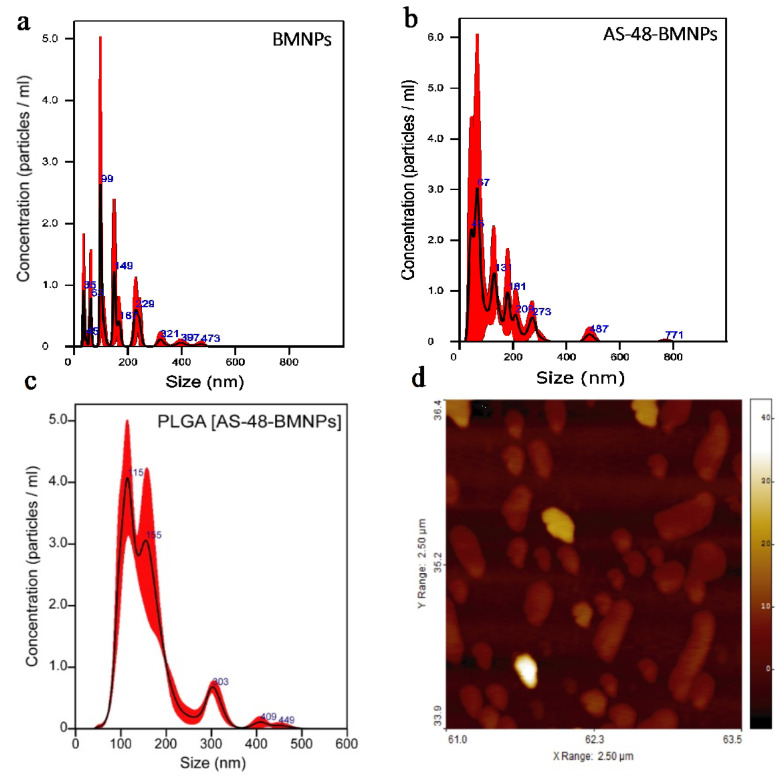
NTA size distribution of (**a**) BMNPs, (**b**) AS-48-BMNPs, and (**c**) PLGA[AS-48-BMNPs]. (**d**) AFM analysis of PLGA[AS-48-BMNPs].

**Figure 4 pharmaceutics-14-02744-f004:**
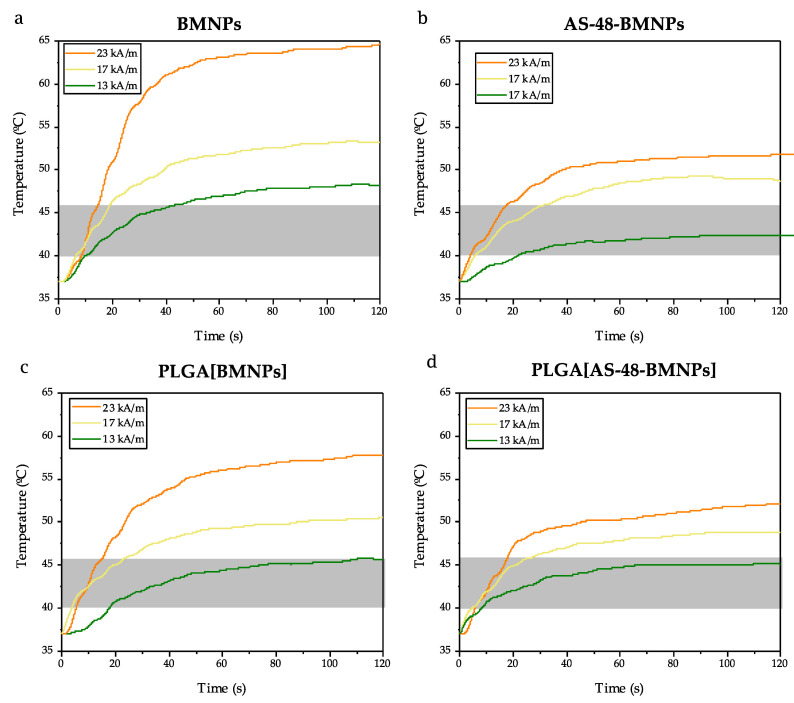
Time evolution of the temperature increase caused by (**a**) BMNPS, (**b**) AS-48-BMNPs, (**c**) PLGA[BMNPs], and (**d**) PLGA[AS-48-BMNPs] following exposure to AMF of different magnetic field strengths (13, 17, 23 kA/m) at a fixed frequency of 120 kHz. The transparent grey band shows the therapeutic temperature.

**Figure 5 pharmaceutics-14-02744-f005:**
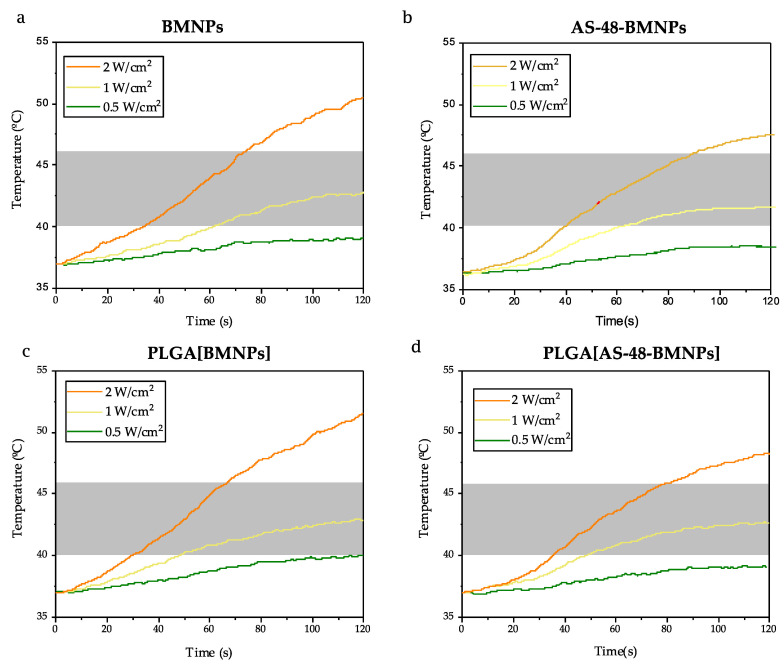
Time evolution of the temperature increase caused by (**a**) BMNPS, (**b**) AS-48-BMNPs, (**c**) PLGA[BMNPs], and (**d**) PLGA[AS-48-BMNPs] following exposure to different laser power densities (0.5, 1, 2 W/cm^2^). The transparent grey band shows the therapeutic temperature.

**Figure 6 pharmaceutics-14-02744-f006:**
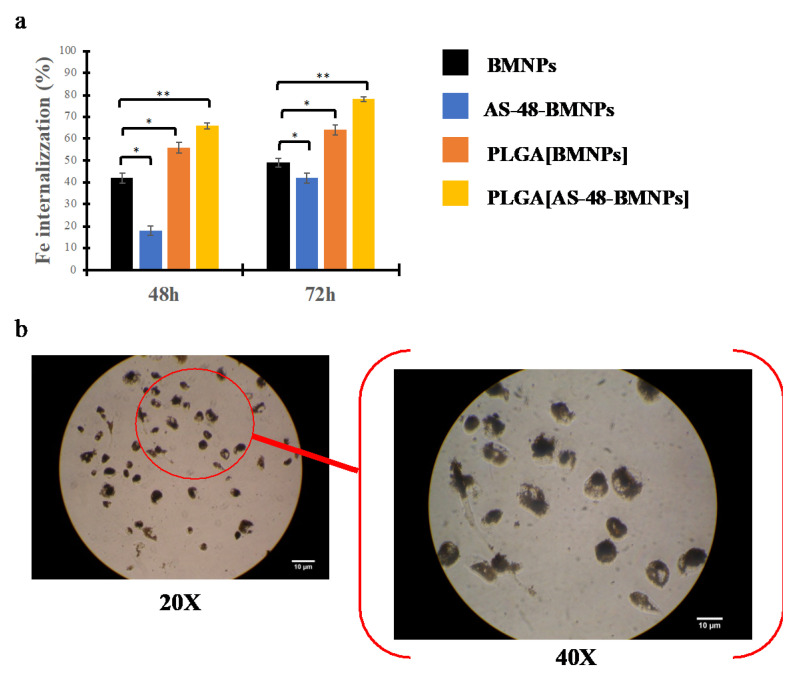
(**a**) Quantitative analysis of BMNPs by measuring the Fe content inside the macrophages after 48 and 72 h in TH1-P cell line following treatment with BMNPs (229 μg/mL), AS-48-BMNPs (229 μg/mL BMNPs and 32 μg/mL AS-48), PLGA[BMNPs] (457 μg/mL PLGA, 229 μg/mL BMNPs), PLGA[AS-48-BMNPs] (457 μg/mL PLGA, 229 μg/mL BMNPs and 32 μg/mL AS-48). (**b**) Brightfield microscopy of THP-1 cells treated 72 h with PLGA[AS-48-BMNPs] * *p* ≤ 0.05, ** *p* ≤ 0.001.

**Figure 7 pharmaceutics-14-02744-f007:**
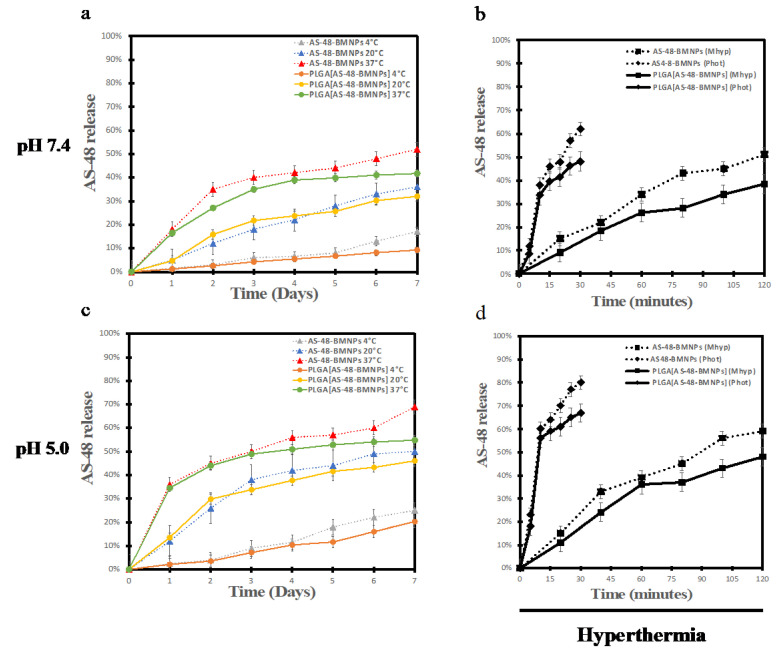
AS-48 released data from PLGA[AS-48-BMNPs] and AS-48-BMNPs in PBS pH 7.2–7.4 (**a**) and in citrate pH 5.0 (**c**) at different temperatures (4, 20, and 37 °C). AS-48 released data from the same nano-formulations following magnetic hyperthermia (Mhyp) and photothermia (Phot) applied for 120 and 30 min, respectively, at pH 7.2–7.4 (**b**) and pH 4.5–5.0 (**d**).

**Figure 8 pharmaceutics-14-02744-f008:**
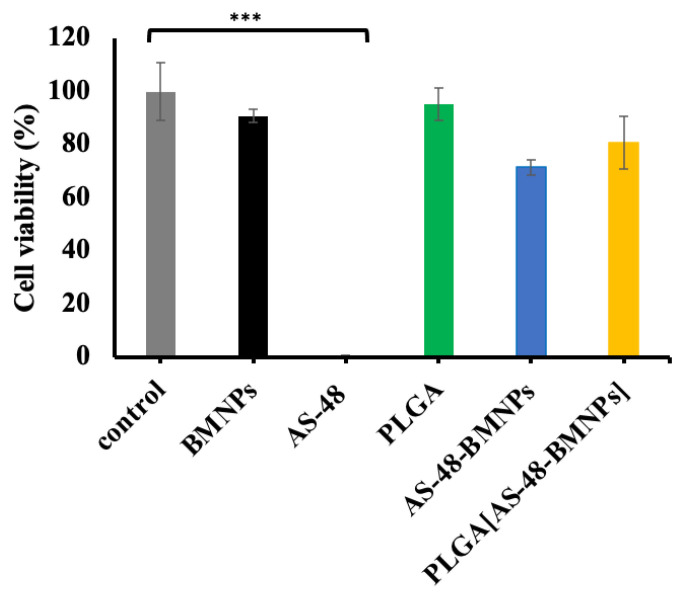
Viability assay in TH1-P cell line following treatment with culture medium (control), BMNPs (229 μg/mL), AS-48 (32 μg/mL), PLGA (457 μg/mL), AS-48-BMNPs (229 μg/mL BMNPs and 32 μg/mL AS-48), PLGA[AS-48-BMNPs] (457 μg/mL PLGA, 229 μg/mL BMNPs and 32 μg/mL AS-48). *** *p* ≤ 0.0001 when compared to control values.

**Table 1 pharmaceutics-14-02744-t001:** Summary of the nano-formulations prepared and analyzed in the present study.

Nanomaterial	Description	Preparation Method
BMNPs	Biomimetic magnetic nanoparticles	Protein-mediated coprecipitation
AS-48-BMNPs	AS-48 immobilized on BMNPs	Ionic coupling
PLGA	Empty PLGA nanoparticles	Single emulsion
PLGA [BMNPs]	PLGA nanoparticles bearing BMNPs	Single emulsion
PLGA [AS-48-BMNPs]	PLGA nanoparticles bearing AS-48-BMNPs	Single emulsion

**Table 2 pharmaceutics-14-02744-t002:** NTA data compared to the AFM diameter. NTA average size represents the average obtained from three independent measures while the NTA mode size is the value most frequently detected. The values correspond to three independent experiments for which standard deviations were calculated.

Sample	NTA Average Size (nm)	NTA Mode Size (nm)	AFM Average Diameter (nm)
BMNPs	164 ± 87	107 ± 15	77 ± 35
AS-48-BMNPs	128 ± 100	67 ± 12	91 ± 50
PLGA	131 ± 26	117 ± 5	182 ± 29
PLGA [BMNPs]	173 ± 72	114 ± 9	201 ± 75
PLGA [AS-48-BMNPs]	160 ± 68	115 ± 7	228 ± 93

**Table 3 pharmaceutics-14-02744-t003:** ζ-Potential (mV) values at physiological and acidic conditions, pH 7.4 and pH 5, respectively. The data correspond to the average of three independent experiments for which the standard deviation has been calculated.

Sample	ζ-Potential (mV) pH 7.4	ζ-Potential (mV) pH 5
BMNPs	−32 ± 6	−6 ± 5
AS-48-BMNPs	−15 ± 4	+10 ± 5
PLGA	−10 ± 3	−1 ± 0.2
PLGA [BMNPs]	−21 ± 6	−3 ± 1
PLGA [AS-48-BMNPs]	−18 ± 4	+4 ± 2

## Data Availability

Not applicable.

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
