# Peer review of "Embedding Biomimetic Magnetic Nanoparticles Coupled with Peptide AS-48 into PLGA to Treat Intracellular Pathogens"

_pharmaceutics, 2022, doi:10.3390/pharmaceutics14122744_

Round 1

Reviewer 1 Report

The manuscript written by S.C. Gaglio et al. is focused on the design of a nanoformulation that allows the combination of directed chemotherapy and local hyperthermia and that could be potentially used against intracellular pathogens. The obtained results are very valuable mainly due to cytotoxicity experiments and their cellular uptake. I am sure that the subject of this contribution in itself is interesting for Pharmaceutics readers, and I have no problem with recommending this article for publication in Pharmaceutics.

Reviewer 2 Report

Re: Embedding biomimetic magnetic nanoparticles, coupled with 2 AS-48 peptide, into PLGA for the treatment of intracellular 3 pathogens”

In this work, the authors prepared a [PLGA-AS-48-BMNPs] nano-assembly by enclosing the AS-48 labelled biomimetic magnetic nanoparticles into PLGA nanoparticles. The drug load can be released in physiological conditions or accelerated at magnetic/photo hyperthermia conditions. This is a follow-up work from their previous works (cited in this manuscript) in AS-48-BMNPs and [TAT-PLGA-DOXO-BMNPs]. This manuscript is well written, and conclusions are supported by experimental data. However, a similar proof of concept work has been published in Pharmaceutics (2021, 13, 1168) and there is no new progress in this work. This work is not recommended for publication in Pharmaceutics without a major revision including the test in a real infection model.

Some minor issues need to be addressed:

11.The releasing rate of AS-48 is temperature dependant. The nano-assembly can be heated either through magnetic hyperthermia or photothermia. Compare Figure 4 and Figure 5, AMF raise the temperature faster than photo irradiation. However, in Figure 7, why the releasing of AS-48 is much faster for photothermia than that of magnetic hyperthermia? 

22. Figure 1 is blurred. The unit for x axis is ‘wave number’, not wavelength.

33. Figure 3.a, missing y axis legend.

44. Format the style of the formulas.

Reviewer 3 Report

In this manuscript, the authors developed an original nanoformulation combining magnetic nanoparticles with a bactericidal protein. The ultimate goal of this nanomaterial would be to treat intracellular bacterial infection such as mycobacterium located inside macrophages that are known to remain stored in macrophages in long-term dormant infections. The manuscript is well written and the data are convincing for the characterization of the nanomaterials as well as for the assay of their cytotoxicity on mammalian cells. However, to really provide a proof-of-concept for the application proposed by the authors, there is a need to provide at least bactericidal assays and the best would be assays showing the capacity the kill intracellular bacterial in infected macrophages, while preserving macrophage viability. Following are more specific comments and corrections to be done before manuscript can be accepted for publication.

There are some typos to correct such as line 348 : ’35. 63, 99’ should be changed for ’35, 63, 99’.

Line 483 : ‘Figure 6b’ should be changed for ‘Figure 7b’.

For the section about cellular uptake starting line 454, I would have preferred that Fe internalization was expressed in a quantitative way more than in % of internalization that remain obscure to me.

The description of the results presented in figure 7 (line 477 to 493) did not stress enough the difference in time scale for the release of AS48 between conditions with or without hyperthermia. Indeed, hyperthermia leads to minute time-scale compared to day time-scale without.

Round 2

Reviewer 2 Report

Re:

Embedding biomimetic magnetic nanoparticles, coupled with 2 AS-48 peptide, into PLGA for the treatment of intracellular 3 pathogens”

The authors have addressed the questions from the last review. The importance for applying this nanoformular for AS-48 delivery has been highlighted in the revision. The time/resource limitations from the authors institute to carry out the in vivo or in vitro model study are acknowledged. This manuscript is well written, and results are supported by experiment data. This work is thus recommended for publication in Pharmaceutics after addressing the questions list below.

1. Compare Figure 4 and Figure 5, AMF raise the temperature faster than photo irradiation. The temperature is monitored for 120s for photothermia and magnetic hyperthermia and the system temperature is almost stabilized within 120s (42.5 ˚C for 1W/cm2 photothemia and 50°C for 23kA/m magnetic hyperthermia). The releasing rate of AS-48 is temperature dependant. While in Figure 7, it is odd that the releasing of AS-48 is much faster for photothermia (releasing condition: 42.5 ˚C for 1W/cm2) than that of magnetic hyperthermia (releasing condition: 50°C for 23kA/m).

2. Figure 1, the unit for wavenumber is cm-1 not nm. Page 13, line 474, “FTIR analysis shows that AS-48 has a peak close to 808 nm 474 (801 nm)”, the unit is cm-1. Please check the whole document for units.
